# Personality links with lifespan in chimpanzees

Drew M Altschul[1,2,3]*, William D Hopkins[4,5], Elizabeth S Herrelko[6,7], Miho Inoue-Murayama[8,9], Tetsuro Matsuzawa[10,11,12], James E King[13], Stephen R Ross[14], Alexander Weiss[1,2]

[1]Department of Psychology, School of Philosophy, Psychology and Language Sciences, The University of Edinburgh, Edinburgh, United Kingdom; [2]Scottish Primate Research Group, United Kingdom; [3]Centre for Cognitive Ageing and Cognitive Epidemiology, Edinburgh, United Kingdom; [4]Neuroscience Institute, Georgia State University, Atlanta, United States; [5]Division of Developmental and Cognitive Neurosciences, Yerkes National Primate Research Center, Atlanta, United States; [6]National Zoological Park, Smithsonian Institution, Washington, United States; [7]Psychology Division, University of Stirling, Stirling, United Kingdom; [8]Wildlife Research Center, Kyoto University, Kyoto, Japan; [9]Wildlife Genome Collaborative Research Group, National Institute for Environmental Studies, Tsukuba, Japan; [10]Institute for Advanced Study, Kyoto University, Kyoto, Japan; [11]Primate Research Institute, Kyoto University, Inuyama, Japan; [12]Japan Monkey Centre, Inuyama, Japan; [13]Department of Psychology, University of Arizona, Tucson, United States; [14]Lester E. Fisher Center for the Study and Conservation of Apes, Lincoln Park Zoo, Chicago, United States

**\*For correspondence:**
dmaltschul@gmail.com

**Competing interests:** The authors declare that no competing interests exist.

**Abstract** Life history strategies for optimizing individual fitness fall on a spectrum between maximizing reproductive efforts and maintaining physical health over time. Strategies across this spectrum are viable and different suites of personality traits evolved to support these strategies. Using data from 538 captive chimpanzees (*Pan troglodytes*) we tested whether any of the dimensions of chimpanzee personality – agreeableness, conscientiousness, dominance, extraversion, neuroticism, and openness – were associated with longevity, an attribute of slow life history strategies that is especially important in primates given their relatively long lives. We found that higher agreeableness was related to longevity in males, with weaker evidence suggesting that higher openness is related to longer life in females. Our results link the literature on human and nonhuman primate survival and suggest that, for males, evolution has favored the protective effects of low aggression and high quality social bonds.
DOI: https://doi.org/10.7554/eLife.33781.001

## Introduction

Life-history theory posits that strategies for increasing individual fitness lay on a continuum that describes an energetic trade-off between maximizing reproductive efforts and maintaining physical health as the organism ages (*Stearns, 1976*). At one end of this continuum are 'r-selected' populations. Individuals within these populations are characterized by early and frequent reproduction, the rapid onset of senescence, and a shorter lifespan. At the other end of this continuum are 'K-selected' populations. Individuals within these populations are characterized by later and less frequent reproduction, but delayed senescence, and a longer lifespan. Both ends of this continuum are viable

**eLife digest** Like humans, animals have distinct personalities. Our close evolutionary cousins chimpanzees even display the same five major personality traits that we do – extraversion, neuroticism, conscientiousness, openness, and agreeableness – as well as a distinct trait, for dominance.

How did these distinct personality traits evolve and persist across different species? Ultimately, each trait must provide some fitness benefits that help the animal to reproduce and pass on the trait to its offspring. Longevity is an important factor in promoting fitness; an animal that lives for longer will have more opportunities to reproduce. Previous work in humans and other animals suggested that some personality traits are associated with a longer life. However, few studies have been large enough to test all major personality traits in both sexes of an animal species.

Altschul et al. used data from a long-term study of 538 captive chimpanzees to investigate possible associations between longevity and personality traits. The personalities of the chimpanzees started being rated between seven and 24 years ago. Since then, 187 of the chimpanzees have died.

Altschul et al. found that different personality traits were associated with longer life in males and females. Male chimpanzees with higher levels of agreeableness – the personality trait characterized by low aggression and positive social interactions such as cooperation – generally lived for longer. Female chimpanzees who were more open to new experiences also appeared to live for longer, but this apparent association may be influenced by age. Like humans, chimpanzees become less open to experiences as they become older.

No other personality traits appear to be linked to lifespan in chimpanzees. However, evidence suggests that conscientiousness and neuroticism can influence lifespan in humans. These two traits may therefore drive uniquely human behaviours that affect health.

The results presented by Altschul et al. suggest that human and ape agreeableness evolved through individuals who were able to achieve higher fitness by living longer. They also provide insights into how important personality and life history are to the health and survival of captive animals. For a fuller understanding of how ape personality evolved, future work should explore longevity and fitness in wild chimpanzees, as well as in our other closest relatives, bonobos.
DOI: https://doi.org/10.7554/eLife.33781.002

fitness strategies, as are, depending upon ecological and social contingences, life history strategies between these extremes. These strategies are supported by behavioral adaptations (*Stearns, 1976*).

Differences in life history strategy have been advanced as one possible explanation for why individuals within populations exhibit stable differences in behavioral, affective, and cognitive dispositions, that is, personality traits (*Dingemanse and Réale, 2005*; *Réale et al., 2010*). A simulation study indicated that this theory is plausible (*Wolf et al., 2007*), and a meta-analysis on studies of boldness, exploration, and aggression in insects, fish, birds, and mammals offered mixed empirical support (*Smith and Blumstein, 2008*). This meta-analysis showed that bolder animals put themselves at greater risk and die at younger ages, but enjoy greater reproductive success than their shyer counterparts, which do not enjoy as many opportunities for copulation, but live longer, and so are able to invest more in their offspring (*Smith and Blumstein, 2008*). Boldness therefore is associated with a 'faster' (*r*-selected) life-history strategy. The findings of the meta-analysis for exploration and aggression were less clear: more aggressive individuals had greater reproductive success than less aggressive individuals, but this was not offset by reduced lifespan; individuals more prone to exploring their environment lived longer than neophobic individuals, but did not experience reduced reproductive success (*Smith and Blumstein, 2008*). Two concurrent reviews showed that, across a range of species, greater boldness, activity, and aggressiveness, and lower sociability and exploration, were associated with a faster life history strategy (*Réale et al., 2010*; *Biro and Stamps, 2008*).

Recent research found evidence that variation in the personality traits of humans and nonhuman primates are also associated with variables related to life history strategies. Studies of humans predominate this literature and, although there are exceptions (e.g., *Alvergne et al., 2010*; *Gurven et al., 2014*), this human literature grew out of personality psychology, health psychology,

and epidemiology. Consequently, these studies did not set out to deliberately test whether personality variation reflected individual differences in life history.

The studies of human personality described above tended to focus on one or more of five traits - extraversion, agreeableness, openness, neuroticism, and conscientiousness - known collectively as the 'Big Five' or 'Five-Factor Model' (*Digman, 1990*). These five traits are operationalized as dimensions onto which several related lower-order traits cluster (*Digman, 1990*). Four of the five human traits correspond to personality traits studied by behavioral ecologists. Extraversion and agreeableness characterize how often and how well humans navigate their social world (*Digman, 1990*). Among other characteristics, extraversion features sociability and activity (*Costa and McCrae, 1995*), which are comparable to the same-named traits studied in behavioral ecology; agreeableness is the opposite of aggressiveness (*Réale et al., 2007*). Openness captures curiosity, originality, and a tendency to find novel ideas and situations appealing (*Digman, 1990*), and corresponds to exploration (*Réale et al., 2007*). Finally, neuroticism is related to fearfulness, vigilance, and emotional reactivity (*Digman, 1990*), and so appears to be the opposite of boldness, that is shyness or timidity (*Réale et al., 2007*). Conscientiousness describes individual differences in self-control, delay of gratification, and thoughtful planning (*Digman, 1990*). Animal analogues of conscientiousness have emerged in a few nonhuman primates, for example chimpanzees (*King and Figueredo, 1997*), and in Asian elephants (*Seltmann et al., 2018*). However, conscientiousness has only recently been operationalized in ways familiar to behavioral ecologists, that is as naturally occurring behaviors or responses to behavioral tests (*Delgado and Sulloway, 2017*; *MacLean et al., 2014*; *Altschul et al., 2017*). In this literature, conscientiousness is often termed 'self-control' (e.g., *MacLean et al., 2014*).

In addition to its focus on the Big Five traits, the life history variables most often examined in the human literature have been health outcomes, especially longevity. Meta-analyses of this extensive literature showed that people who enjoy better health and live longer tend to be higher in agreeableness, extraversion and conscientiousness, and lower in neuroticism (*Strickhouser et al., 2017*; *Roberts et al., 2007*). The explanatory theories emerging from this field posit that health-related behaviors, including diet, mediate relationships between personality and health (*Turiano et al., 2015*; *Graham et al., 2017*). The possibilities that agreeableness, extraversion and conscientiousness are related to a slower life history strategy, and that neuroticism is related to a faster life history strategy, are mostly not considered in this literature.

Studies of personality and life history in nonhuman primates are often narrower in scope than studies of humans. Specifically, they mostly test whether one or more personality traits related to social interactions are associated with health and/or mortality outcomes. This narrow focus is probably attributable to two characteristics of these species. First, nonhuman primates have relatively slow life-history strategies; lifespans are comparatively long and reproductive rates are comparatively low (*Jones, 2011*). Consequently, health and longevity are influential fitness measures in primates, including humans. Second, most primate species live in groups and are highly social (*Napier and Napier, 1967*). To date, whether they use rating and/or coding measures of personality, studies of personality and survival in nonhuman primates have shown that western lowland gorillas (*Weiss et al., 2013*), baboons (*Silk et al., 2010*; *Archie et al., 2014*; *Seyfarth et al., 2012*), and female rhesus macaques (*Brent et al., 2017*) that are higher in sociability live longer. However, a study of female blue monkeys found that the association between sociability and mortality was only true for individuals that had consistent bonds with groupmates (*Thompson and Cords, 2018*).

In addition to the fact that all but one of these studies focus on a narrow set of traits (*Weiss et al., 2013*), studies of primate personality and longevity have focused on a small number of species. In particular, New World monkeys are not represented and only one study was of a species of great ape (*Weiss et al., 2013*), the evolutionary line that includes humans. We wished to expand on what is known about the links between personality traits and life history strategy in nonhuman primates and in humans. To do so we examined these associations in chimpanzees, which are one of our closest living great ape relatives.

The present study was made possible by the existence of a database containing a large sample (*n* = 538) of captive chimpanzees living in zoological parks, research facilities, and sanctuaries located in the United States, the United Kingdom, the Netherlands, Australia, and Japan. Personality in this sample was assessed by ratings on two comparable questionnaires that assessed a wide range of traits. These ratings were made by keepers, researchers, and others who knew and worked with these chimpanzees for considerable lengths of time. Furthermore, the long follow-up times from

when chimpanzees' personalities were assessed to the present (7 to 24 years) meant that there were enough deaths to provide adequate statistical power for detecting associations between personality and mortality. The sample used in this study and the means of measuring personality deserve comment.

There is some disagreement as to whether chimpanzees or bonobos, which are as related to humans as chimpanzees, are the best model for ancestral humans (*Stanford, 2012*; *Sayers et al., 2012*). However, studies using similar personality measures in captive groups of chimpanzees and bonobos have found that the dimensions along which chimpanzee personality traits align themselves (*King and Figueredo, 1997*) are more similar to the human dimensions than are those of bonobos (*Weiss et al., 2015*). Specifically, in addition to a dominance dimension, which reflects competitive prowess, social competence, and fearlessness, that is not present in humans (*King and Figueredo, 1997*; *Murray, 1998*; *Dutton et al., 1997*; *Freeman et al., 2013*; *Weiss et al., 2009*; *Weiss et al., 2007*), chimpanzee personality is defined by five dimensions that resemble the human Big Five. These dimensions have been identified in many studies, including those that measured personality with different questionnaires (*King and Figueredo, 1997*; *Murray, 1998*; *Dutton et al., 1997*; *Freeman et al., 2013*; *Weiss et al., 2009*; *Weiss et al., 2007*; *King et al., 2005*; *Martin, 2005*; *Buirski et al., 1978*) and those that used coded behavioral observations instead of ratings (*Freeman et al., 2013*; *Massen et al., 2013*; *Koski, 2011*; *Vazire et al., 2007*; *Pederson et al., 2005*; *van Hooff, 1970*). In bonobos, questionnaire-based and coding-based methods revealed evidence for human- and chimpanzee-like agreeableness, conscientiousness, and openness dimensions, a dimension like the chimpanzee dominance dimension, and an additional dimension, attentiveness, which is distinct from conscientiousness (*Weiss et al., 2015*; *Staes et al., 2016*). However, these studies find next to no evidence for neuroticism and extraversion. Taken with findings from comparable studies of the other great apes (*Weiss et al., 2006*; *Gold and Maple, 1994*), one plausible scenario is that bonobo personality diverged from that of chimpanzees and the other great apes, including humans.

Some question the use of ratings to measure animal personality given the possibility of anthropomorphic projection (*Uher, 2013*). For studies of nonhuman primates, as noted in the previous paragraph, ratings and behavioral measures yield comparable personality traits. Moreover, a review and meta-analysis found evidence that different raters provide similar ratings, that these measures are heritable, and that they are repeatable (*Freeman and Gosling, 2010*), the latter being most recently demonstrated in ratings taken 35 years apart and made by two independent sets of raters on two different questionnaires (*Weiss et al., 2017*). In addition, the effects of anthropomorphic projection by raters, if present, are minimal (*Weiss et al., 2012*). These just-described findings are probably attributable to the fact that items on most questionnaires do not consist of a single word (typically an adjective), but include behavioral definitions, which limit the degree of subjectivity in interpreting the traits and making ratings (*Uher and Asendorpf, 2008*; *Stevenson-Hinde and Zunz, 1978*).

Another concern that some raise is the use of captive samples. Although they limit the conclusions that we can draw about ancestral humans, by using captive samples one is able to remove many extrinsic sources of mortality, for example predators and infectious diseases. Therefore, captive samples, such as that used in this study, control for potential confounds that might crop up in studies of wild samples. In addition, captive samples are uniquely suited to testing whether the associations between human personality and mortality risk reflect life history strategies followed by individuals apart from links between personality and health-related behaviors that are endemic to human personality studies.

We used these data to test six hypotheses, one for each chimpanzee personality trait. We will first describe the hypotheses for the chimpanzee personality traits of extraversion, agreeableness, openness, and neuroticism, which are closely related to traits studied by behavioral ecologists. We will then describe the hypotheses for conscientiousness and dominance, which were based on literature that we will discuss.

Because sociability and aggressiveness are associated with slower and faster life-history strategies, respectively (*Réale et al., 2010*; *Brent et al., 2017*), we expect that higher extraversion and agreeableness will be related to longer life. In nonhumans, lower boldness is related to a slower life-history strategy. In humans, although overall neuroticism is associated with poorer health and a shorter lifespan, aspects of neuroticism related to worry and vigilance, key characteristics related to lower boldness (*Réale et al., 2007*), are associated with *better* health and a *longer* lifespan

(*Gale et al., 2017*; *Weston and Jackson, 2018*). We thus expect that neuroticism should be associated with a longer life-span. Exploration, in animals, is linked to some characteristics of a slower life history, and so we expect that openness in chimpanzees will be associated with longer life.

We expect that conscientiousness will be related to a slower life history, and so longer life. This expectation was based on the above-described finding that humans who are higher in conscientiousness enjoy better health and live longer. If we do not find such an association, it would suggest that the association between conscientiousness and better health in humans may be attributable to human-specific health behaviors, such as exercising, that are related to higher conscientiousness and lead to individuals being healthier (*Turiano et al., 2015*). Our basis for this interpretation of these results stems from the fact that captive chimpanzees do not have many (if any) opportunities to control their health, which is in fact maintained by humans.

Finally, among primates, social standing is related to physiological stress responses (*Sapolsky, 2005*) and high dominance is associated with higher stress, as well as faster, energetically intense growth in chimpanzees (*Pusey et al., 1997*). High-ranking individuals also mate more frequently and dominate resources to support their growth and reproductive efforts (*Ellis, 1995*). Higher rank in chimpanzees, therefore, is associated with a faster life history strategy. Because ratings on traits such as dominance in chimpanzees and other primates are related to rank, including in the wild (*Buirski et al., 1978*), we expected that dominance would be related to a shorter lifespan.

## Results

### Comparing captive and wild chimpanzee mortality

During the follow-up period, 187 chimpanzees died. A Kaplan-Meier plot (*Figure 1*) shows survival functions for our sample and a wild sample (*Bronikowski et al., 2011*). Unlike wild chimpanzee populations in which infant mortality is high, captive chimpanzee populations have strikingly reduced infant mortality, live longer, and display accelerated mortality in older ages. These results show that captive chimpanzees benefit from protection against extrinsic sources of mortality, for example shelter from elements and predators, good health care, and abundant food.

### Associations between personality and age

Inspection of the six chimpanzee personality dimensions (*Figure 2*), as well as prior studies (*King et al., 2008*) indicate that personalities change as individuals age, making it possible that an association between personality and longer life might be confounded. This is not necessarily undesirable, as it indicates that personality and lifespan are linked, but to be conservative, we modeled and therefore controlled for potential confounds between age and personality scores. We fitted generalized additive models (GAMs) for each personality dimension, regressing personality ratings on the age at which the individual was rated.

The GAM regression lines for each model are plotted against the personality data in *Figure 2— figure supplements 1 through 6*. Curvilinear associations were presented between age and personality for all dimensions except neuroticism, where only a linear relationship was present.

Because personality does change over time, some of the raw personality score variance could be attributed to rating age variance. Alternative, adjusted personality scores were therefore calculated as residuals from the regression function of each GAM. In the subsequent analyses, adjusted scores were fitted as predictors in separate survival models from the raw scores.

### Decision tree survival models

We fit decision trees to test whether sex, origin (wild-born or other), or any personality dimensions were related to longevity. A conditional inference survival tree procedurally determined that among males, higher agreeableness was associated with longer survival (*Figure 3*). Specifically, males with agreeableness scores less than 0.063 standard deviations below the mean were at significantly higher risk than other males (p<0.027). These results held for the age-adjusted agreeableness scores as well.

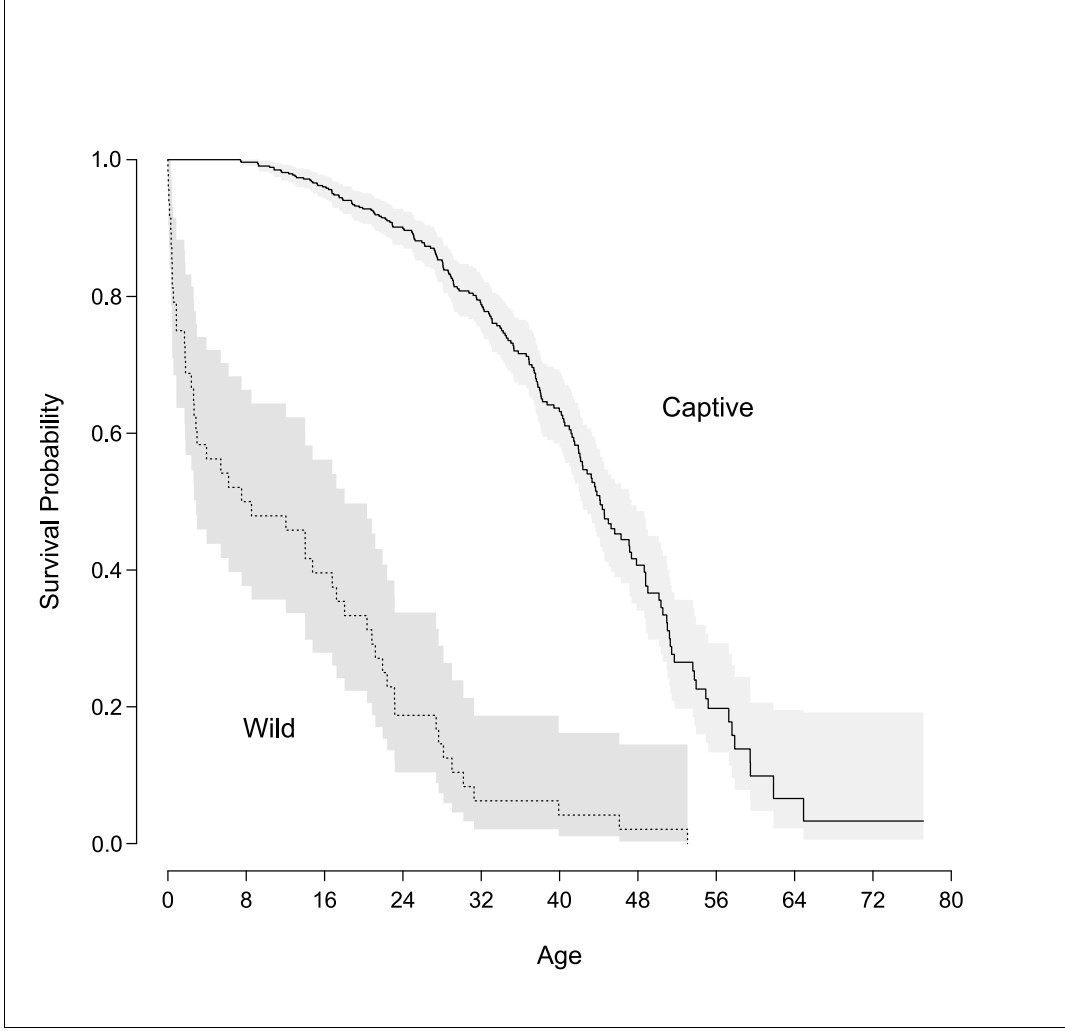

**Figure 1.** Survival curves of captive and wild chimpanzees. Lines indicate survival probability of each group over the lifespan. The solid lines represent the captive population used in this study and the dashed line corresponds to a wild group (*Bronikowski et al., 2011*). The shaded areas indicated the 95% confidence region for reach group.

DOI: https://doi.org/10.7554/eLife.33781.003

## Weighted parametric hazard regression models

The association between agreeableness and survival in males was confirmed with parametric hazards modeling: in a AIC weighted model including all covariates and frailty effects, the hazard ratio for males was 0.66 (95% CI: 0.49 – 0.89) per standard deviation increase, and in a model where we adjusted personality scores to control for age, the hazard ratio associated with a standard deviation increase was 0.61 (95% CI: 0.42 – 0.89). In the models of only females, a positive association between openness and survival was also revealed with a hazard ratio of 0.77 (95% CI: 0.59 – 0.99) for unadjusted scores, but the association was not significant when we used the adjusted openness scores. Higher openness in males was not related to living longer nor was higher agreeableness in females (*Table 1* presents a full description of the AIC weighted models). For a subset of the sample, more detailed rearing data were available, but survival analyses did not find any association between rearing conditions or origin and longevity (Table S1). A complete description of all survival analyses is available in the supporting information.

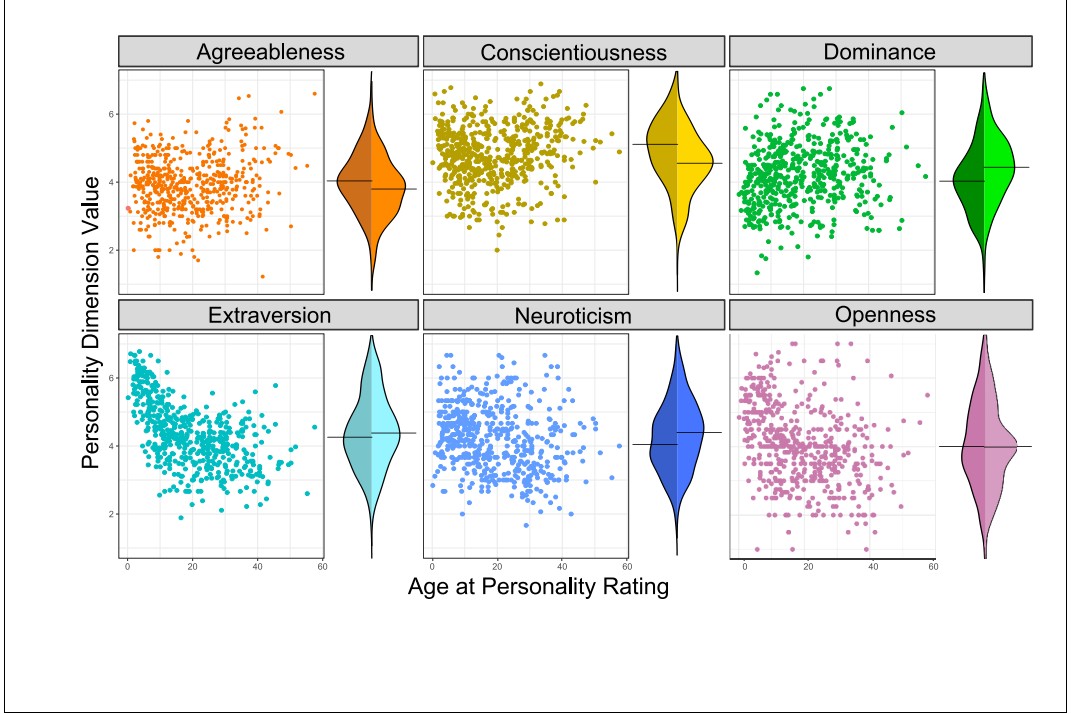

**Figure 2.** Personality's relationship with age and sex. Each panel shows the personality scores of a specific dimension for all individuals in a scatterplot against age on the left, and on the right with bean plots showing the distribution of scores split by sex (females are on the left, males on the right). Relationships between age and each personality dimensions are illustrated in the figure supplements.

DOI: https://doi.org/10.7554/eLife.33781.004

The following figure supplements are available for figure 2:

**Figure supplement 1.** Generalized additive model of dominance and age at personality rating.
DOI: https://doi.org/10.7554/eLife.33781.005

**Figure supplement 2.** Generalized additive model of extraversion and age at personality rating.
DOI: https://doi.org/10.7554/eLife.33781.006

**Figure supplement 3.** Generalized additive model of conscientiousness and age at personality rating.
DOI: https://doi.org/10.7554/eLife.33781.007

**Figure supplement 4.** Generalized additive model of agreeableness and age at personality rating.
DOI: https://doi.org/10.7554/eLife.33781.008

**Figure supplement 5.** Generalized additive model of neuroticism and age at personality rating.
DOI: https://doi.org/10.7554/eLife.33781.009

**Figure supplement 6.** Generalized additive model of openness and age at personality rating.
DOI: https://doi.org/10.7554/eLife.33781.010

## Discussion

We found a clear pattern of relationships between personality and longevity in these data: among males, higher agreeableness was associated with longer life, even when agreeableness was adjusted for age. In other words, long-living captive male chimpanzees are those who engage in positive social interactions characterized by cooperation, geniality, and being protective. These findings match our prediction, although we did not necessarily expect to find the association only in males. However, this finding is consistent with the literature: in wild chimpanzees, male coalitionary aggression towards conspecifics is associated with greater chances of siring offspring (*Gilby et al., 2013*). Agreeableness, the opposite of aggression, ought to lie on the other end of the life-history spectrum, and be associated with longer life, as we found. More agreeable males may adopt a more cooperative dominance style (*Foster et al., 2009*), ultimately allowing for fewer, but more consistent reproductive opportunities over the course of a long life.

We were surprised to find no association between extraversion and longevity. Studies in monkeys (*Silk et al., 2010*; *Seyfarth et al., 2012*; *Brent et al., 2017*) have shown positive, protective

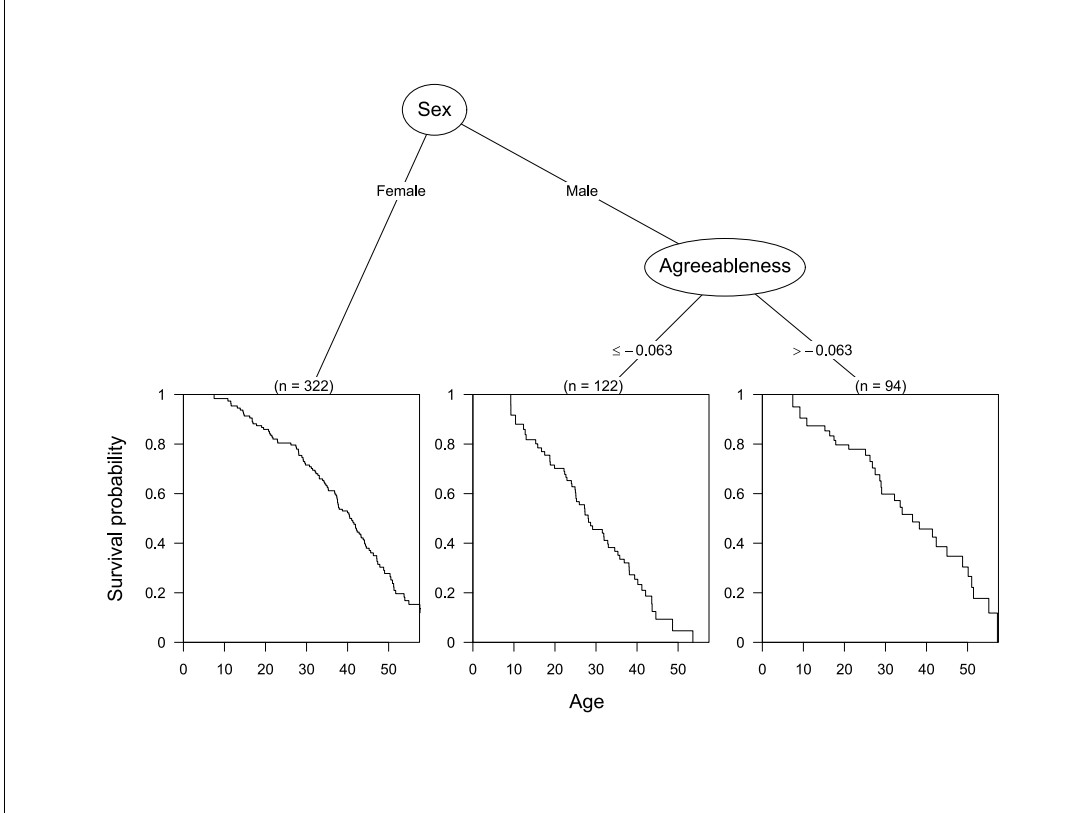

**Figure 3.** Conditional inference tree diagram indicating variables influencing survival. Bottom panes indicate the survival curves of and number of chimpanzees in each sub-group. Sub-groups were split based on the growth of the tree and decision criteria are indicated below each node. Splits in numeric variables (e.g. agreeableness) are by standard deviations.

DOI: https://doi.org/10.7554/eLife.33781.011

relationships with extraversion. Of note, a positive association between extraversion and longevity was found in a study of gorillas that were also kept in captivity and assessed for personality by means of ratings (*Weiss et al., 2013*). Like their close chimpanzee cousins, captive gorillas show evidence for strong age-related declines in extraversion (*Kuhar et al., 2006*), yet extraversion was still associated with longevity. However, high sociability among primates does not support longevity in all circumstances (*Thompson and Cords, 2018*). The remaining difference between gorillas and chimpanzees that could explain our null findings for extraversion lies in the mating systems of these species. Specifically, gorillas have strict harems where one or two males have exclusive sexual access to multiple mature females (*Harcourt et al., 1981*). Chimpanzees, on the other hand, have a promiscuous mating system (*Tutin, 1979*).

There was no association between longevity and conscientiousness. It is possible that this finding reflects our captive sample in which the extrinsic benefits of being higher in conscientiousness have been removed. For instance, although chimpanzees are known to self-medicate using plants in the wild (*Huffman and Wrangham, 1994*), and while conscientious chimpanzees in captivity are more diligent (*Altschul et al., 2017*), individuals have no resources to use for self-medication in captivity. Our results thus suggest that the associations commonly found between conscientiousness and longevity in human is not related to intrinsic characteristics of the organism, but to the health-related behaviors associated with this trait (*Turiano et al., 2015*).

Females that were higher in openness lived longer, but the effect was not present when we corrected for confounding by age of rating. This is due to the strong curvilinear relationship between age and openness (*Figure 2*). Younger chimpanzees were much higher in openness and there was an association between lower openness and age, a limitation we might have missed had our sample

**Table 1.** Weighted survival model estimates of personality and demographic variables related to longevity.

Values are model averaged parameter estimates and unconditional confidence intervals calculated from estimates shown in Supplementary Table 4.

| | Unadjusted | | Adjusted for age | |
|---|---|---|---|---|
| Variable | Hazard Ratio | 95% C.I. | Hazard Ratio | 95% C.I. |
| Male (n = 216) | | | | |
| Wild-born | 1.40 | [0.68, 2.90] | 1.35 | [0.66, 2.74] |
| Agreeableness | **0.66** | **[0.49, 0.89]** | **0.61** | **[0.42, 0.89]** |
| Dominance | 0.98 | [0.74, 1.29] | 0.99 | [0.72, 1.37] |
| Extraversion | 1.04 | [0.71, 1.51] | 1.01 | [0.65, 1.57] |
| Conscientiousness | 1.11 | [0.78, 1.58] | 1.19 | [0.79, 1.81] |
| Neuroticism | 0.91 | [0.66, 1.25] | 0.93 | [0.66, 1.31] |
| Openness | 1.09 | [0.76, 1.55] | 1.06 | [0.78, 1.46] |
| Female (n = 322) | | | | |
| Wild-born | 1.16 | [0.72, 1.85] | 1.17 | [0.73, 1.87] |
| Agreeableness | 1.12 | [0.83, 1.50] | 1.24 | [0.84, 1.82] |
| Dominance | 1.04 | [0.83, 1.30] | 1.05 | [0.82, 1.35] |
| Extraversion | 1.15 | [0.80, 1.67] | 1.02 | [0.66, 1.57] |
| Conscientiousness | 1.01 | [0.76, 1.34] | 0.98 | [0.70, 1.38] |
| Neuroticism | 0.93 | [0.73, 1.17] | 0.93 | [0.72, 1.19] |
| Openness | **0.77** | **[0.59, 0.99]** | 0.82 | [0.66, 1.02] |

DOI: https://doi.org/10.7554/eLife.33781.012

been smaller. It is therefore impossible for us to conclude whether there is a protective association between openness and longevity in females or whether lower openness was a proxy for age.

Low boldness resembles one aspect of human neuroticism that is related to a longer lifespan, and so we predicted that neuroticism would be associated with greater longevity. However, we found no association in either direction. The absence of any effect of neuroticism in chimpanzees may be attributable to the fact that the health-harming and health-benefitting roles of neuroticism are, like conscientiousness, mediated by health behaviors, as well as the environment. For example, people who are higher in neuroticism tend to smoke, and this behavior explains some of the relationship between neuroticism and shorter lifespans (*Graham et al., 2017*). On the other hand, after the onset of certain diseases, some high neuroticism individuals are more likely to stop smoking (*Weston and Jackson, 2018*). Smoking does not explain the entire association in humans, however, as high neuroticism is also associated with greater reactivity to stressors (*Chapman et al., 2011*) and energetically expensive physiological responses (*Réale et al., 2010*), which could offset potential benefits of slow life-history benefits from neuroticism. Moreover, with the absence of predators in captivity benefits of vigilance would be reduced if not entirely eliminated, as danger and risks to health from agonistic social encounters remain.

Dominance, and the degree to which captive chimpanzees are characterized by their competitive prowess and fearlessness, and, consequently, the ability to enjoy the spoils of rank, had no bearing on how long individuals lived. In chimpanzees specifically, high-ranking individuals are generally less stressed (*Goymann and Wingfield, 2004*), but when the hierarchy is destabilized, high-ranking individuals become more stressed, and instability and reorganization can be common in wild chimpanzee groups (*Muller and Mitani, 2005*). Dominance may not play a major role in influencing longevity in captive populations because fission-fusion dynamics are not in play to the same extent as in the wild, thus group stability will be greater, and stressful disruption will be reduced. Moreover, in captivity there is less need for chimpanzees to compete with one another for resources, so traits such as dominance, that are related to rank, may not be related to mortality in this sort of environment.

This study had several limitations. Our data did not have measures of social variables like rank or social network, or psychological variables like intelligence. These chimpanzees lived exclusively in captive environments, which limits our ability to make evolutionary inferences regarding the associations between personality and survival. However, our captive sample was also a strength as it allowed us to identify extrinsic influencers that would be eliminated by captive environments and test novel hypotheses about the relationships between personality and life-history strategies in chimpanzees.

Our study also examined only a single species. More generally, future studies that incorporate multiple primate species could utilize phylogenetic approaches, which consider the importance of species differences in social organization and ecology (*MacLean et al., 2012*; *Cornwell and Nakagawa, 2017*). Phylogenetic analyses could allow researchers to identify which specific species differences moderate relationships between certain personality traits and measures of health and survival, as well as reproductive success and fitness more broadly.

The present study is a reminder of the complex, multifaceted nature of personality and sex, social relationships and the life course in chimpanzees. It also shows how studying the personality of our biological kin reveals that, as in humans, it is not the quantity of social relationships that matters, but the quality.

## Materials and methods

### Sample and experimental design

All research reported in this study was non-invasive. The research complied with the regulations and guidelines prescribed by The University of Edinburgh and the participating zoos, research institutes, and sanctuaries.

556 chimpanzees were assessed for personality between 1993 and 2010. Eighteen chimpanzees had to be removed from the sample due to incompatibilities with the study design, either because personality was assessed after death or because a veterinary staff member requested the individual not be analyzed and mortality data were thus withheld. Of the 538 remaining chimpanzees, 175 came from zoos in the United States, 164 came from the Yerkes National Primate Research Center (also in the United States), 156 came from zoos, a sanctuary, and two research centers in Japan, 21 came from the Taronga Zoo in Australia, 11 came from the Beekse Bergen Safaripark in the Netherlands, and 11 came from the Edinburgh Zoo in the United Kingdom.

Vital status was recorded throughout 2016 and 2017, yielding follow-up times ranging from 7 to 24 years, which is approximately equivalent to 10 to 36 human years (*Napier and Napier, 1967*). A total of 187 chimpanzees died during the follow-up period. As is standard in studies that seek to identify mortality risk factors, our analytic approach treated the remaining 353 chimpanzees as right-censored at the date that mortality data were gathered for that group. 336 individuals were known to be alive at the time of data collection, and 17 individuals were lost to follow-up and censored at the date of their last known record. All records were also left-truncated, beginning each record at the age at which the individual was assessed for personality.

### Personality assessments

Fifty-four items comprising a trait name, for example 'Fearful' and a one to three sentence behavioral description, for example 'Subject reacts excessively to real or imagined threats by displaying behaviors such as screaming, grimacing, running away or other signs of anxiety or distress.' were developed to assess the personalities of the chimpanzees (*King and Figueredo, 1997*; *Weiss et al., 2009*), Between 1993 and 2005, 43 of these items were used to assess the personalities of chimpanzees in the American zoos, the Taronga Zoo, and chimpanzees living at the Yerkes National Primate Research Center (*King and Figueredo, 1997*; *Weiss et al., 2007*). Starting in 2007, all 54 items were used to assess the personality of the chimpanzees living in Japan (*Weiss et al., 2009*), the Netherlands (*Herrelko, 2011*), and at the Edinburgh Zoo (*Herrelko et al., 2012*). The distributions of all six chimpanzee personality dimensions split by sex are shown in *Figure 2*.

The personalities of the chimpanzees in this study were assessed via ratings on these items by multiple keepers and researchers who knew the individual chimpanzees, sometimes for decades (*King and Figueredo, 1997*; *Weiss et al., 2009*; *Weiss et al., 2007*). In addition to showing that the

interrater reliabilities are comparable to those found in human studies of personality, previous studies have shown that chimpanzee personality, measured this way, yields measures that are more reliable than behavioral codings (*Vazire et al., 2007*), that are heritable (*Weiss et al., 2000*; *Wilson et al., 2017*; *Latzman et al., 2015a*) and stable over time (*King et al., 2008*), and that generalize across samples (*Weiss et al., 2009*; *Weiss et al., 2007*; *King et al., 2005*), and are not adversely affected by anthropomorphic attributions on the part of raters (*Weiss et al., 2012*), Finally, these measures have been related to observed behaviors (*Pederson et al., 2005*), differences in brain morphology (*Latzman et al., 2015b*; *Blatchley and Hopkins, 2010*), and genetic polymorphisms (*Wilson et al., 2017*; *Hong et al., 2011*; *Hopkins et al., 2012*).

## Generalized additive models

To adjust for confounding in the personality variables brought on by changes with age, we fit GAMs modeling the relationship between age at assessment and each personality variable (*Wood, 2006*). GAMs are an extension to linear models that allow the input data to 'suggest' non-linearities (*Hastie, 2017*) as opposed to requiring researchers to manually specify them, by, for example, adding a quadratic term to a model formula. To avoid overfitting, non-parametric transformations penalize roughness in the transformation function creating terms aptly called 'smooths' (*Faraway, 2016*). For our smooths, we used thin plate regression splines with a basis dimension ($k$) of 20. The basis dimension was verified as being acceptable using internal package functions; varying $k$ did not alter any model fits. GAMs are difficult to interpret mathematically, but visually intuitive, so each GAM is described by its line of best fit, drawn in *Figure 2—figure supplements 1 through 6*. GAMs generate residuals like other regression models, thus, bivariate GAMs are a powerful method for identifying and controlling for the effects of confounders (*Benedetti and Abrahamowicz, 2004*).

## Survival analyses

To be conservative, our survival models included all six personality scores. We also included sex and origin (whether the individual was born in the wild or not) as controls.

We used decision-tree analyses to identify associations between personality and longevity. Parametric and semi-parametric survival regression models force a specific link between variables and outcome, but decision trees do not impose any such assumptions; trees are able to automatically identify meaningful variables and even some interactions without prior specification (*Bou-Hamad et al., 2011*). Survival trees in particular have advantages over other techniques. In simulation studies of left-truncated right-censored decision trees with data much like ours, that is a large sample ($N > 500$) with many censored observations (>50%), conditional inference trees identified the correct predictors 94% and 93% of the time, respectively (*Fu and Simonoff, 2016*). This method can handle binary and continuous variables and is robust to the effects of time-dependent covariates, such as our chimpanzees' personality dimensions, which could be confounded with age at rating.

We grew trees with both unadjusted and adjusted covariates. Adjusted covariates were residualized versions drawn from the GAMs used earlier to model the effects of age on personality. Using adjusted covariates had no meaningful effect on the conditional inference analysis; the tree grown was identical.

We validated our decision-tree analyses with fully parametric hazard regression models. We followed an information theoretical approach which allowed us to pool and average model estimates across a wide-range of possible choices of error distribution and variables to include (*Burnham et al., 2011*). We first built two sets of models, again, with unadjusted covariates and without adjusted covariates. Adjustment creates a different, alternative dataset which cannot be directly compared to the unadjusted data, so our evaluations of these models were necessarily kept separate. The linking distributions we used included the Weibull, log-logistic, Gompertz (*Klein and Moeschberger, 2005*), and semi-parametric splines survival functions (*Goodman et al., 2011*). There were no convergence issues and all splines were fit with 12 knots and κ = 10,000. The hazard models were fit with Gamma distributed frailty (random) effects to control for any influence that the different sample groups might have on survival, and estimated both jointly and separately by sex (Table S2 and *Table 1*, respectively). We also built models including and excluding the demographic covariates of sex and origin. No variation in specification affected our results (Tables S3 & S4).

## Acknowledgments

We would like to thank Sara Brice, Donald Gow, Lydia Hopper, Tom Booth, Ian Deary, Vanessa Wilson, and the Great Ape Information Network. This research was partly conducted at The University of Edinburgh Centre for Cognitive Ageing and Cognitive Epidemiology, part of the cross council Lifelong Health and Wellbeing Initiative. Funding from the Biotechnology and Biological Sciences Research Council (BBSRC), Economic and Social Research Council (ESRC) and Medical Research Council (MRC) is gratefully acknowledged.

## Additional information

### Funding

| Funder | Grant reference number | Author |
|---|---|---|
| Japan Society for the Promotion of Science | Grant for Scientific Research ( 25118005,25290082) and Development Fund (D-1007) | Miho Inoue-Murayama |
| Kyoto University | Supporting program for interaction-based initiative team studies (SPIRITS) | Miho Inoue-Murayama |
| Ministry of Education, Culture, Sports, Science, and Technology | 16H06283 | Tetsuro Matsuzawa |
| Medical Research Council | Grant to the Centre for Cognitive Ageing and Cognitive Epidemiology (MR/K026992/1) | Drew Altschul |
| Daiwa Anglo-Japanese Foundation | Small Grant (6515/6818) | Alexander Weiss |
| University Of Edinburgh | Development Trust Small Project Grant | Alexander Weiss |
| National Institutes of Health | Grants to the Yerkes Primate Research Center (NS-36605,NS-42867,RR 00165) | William Donald Hopkins |
| Ministry of Education, Culture, Sports, Science, and Technology | Grant to Scientific Research (B) (18310152) (21310150) | Miho Inoue-Murayama |
| Leo S. Guthman Fund | | Stephen Ross |
| Ministry of Education, Culture, Sports, Science, and Technology | Leading Graduate Program PWS(U04) | Tetsuro Matsuzawa |
| Japan Society for the Promotion of Science | Core-to-core CCSN | Tetsuro Matsuzawa |

The funders had no role in study design, data collection and interpretation, or the decision to submit the work for publication.

### Author contributions

Drew M Altschul, Conceptualization, Data curation, Software, Formal analysis, Validation, Investigation, Visualization, Methodology, Writing—original draft, Project administration, Writing—review and editing; William D Hopkins, James E King, Conceptualization, Resources, Data curation, Funding acquisition, Investigation, Writing—original draft, Project administration; Elizabeth S Herrelko, Resources, Data curation, Investigation, Writing—original draft, Writing—review and editing; Miho Inoue-Murayama, Resources, Data curation, Funding acquisition, Investigation, Writing—original draft, Project administration; Tetsuro Matsuzawa, Resources, Data curation, Funding acquisition, Investigation, Writing—original draft; Stephen R Ross, Conceptualization, Resources, Data curation, Investigation, Writing—original draft, Project administration; Alexander Weiss, Conceptualization,

Resources, Data curation, Software, Formal analysis, Supervision, Funding acquisition, Investigation, Methodology, Writing—original draft, Project administration, Writing—review and editing

**Author ORCIDs**
Drew M Altschul (iD) http://orcid.org/0000-0001-7053-4209
Alexander Weiss (iD) http://orcid.org/0000-0002-9125-1555

### Ethics

Animal experimentation: All of the research reported in this study was noninvasive. The research in this study complied with the regulations and guidelines prescribed by the University of Edinburgh Biological Services' Animal Welfare and Ethical Review Committee (AWERB no. OS04-14) and the participating research institutes (YNPRC IACUC protocol YER-4000125-ENTRPR-A), sanctuaries and zoos (accredited by the Association of Zoos and Aquariums) that opted into the research. American Psychological Association guidelines for the ethical treatment of animals were adhered to during all aspects of this study. The Chimpanzee Species Survival Plan endorsed this research on 27 March 2015.

### Decision letter and Author response

Decision letter https://doi.org/10.7554/eLife.33781.021
Author response https://doi.org/10.7554/eLife.33781.022

## Additional files

### Supplementary files

• Supplementary file 1. Containing Tables S1 – S4.
DOI: https://doi.org/10.7554/eLife.33781.013
• Source code 1. R code for data processing, plotting, and analyses.
DOI: https://doi.org/10.7554/eLife.33781.014
• Transparent reporting form
DOI: https://doi.org/10.7554/eLife.33781.015

### Data availability

All analysed data on sex, origin, rearing (in a subset), age at follow-up, age at rating, and personality scores have been deposited with Dryad. https://doi.org/10.5061/dryad.7hq7pc7

The following dataset was generated:

| Author(s) | Year | Dataset title | Dataset URL | Database, license, and accessibility information |
|---|---|---|---|---|
| Altschul DM, Hopkins WD, Herrelko ES, Inoue-Murayama M, Matsuzawa T, King JE, Ross SR, Weiss A | 2018 | Data from: Personality and longevity in captive chimpanzees | https://doi.org/10.5061/dryad.7hq7pc7 | Available at Dryad Digital Repository under a CC0 Public Domain Dedication |

The following previously published dataset was used:

| Author(s) | Year | Dataset title | Dataset URL | Database, license, and accessibility information |
|---|---|---|---|---|
| Bronikowski AM, Altmann J, Brockman DK, Cords M, Fedigan LM, Pusey A | 2011 | Data from: Aging in the natural world: comparative data reveal similar mortality patterns across primates | http://dx.doi.org/10.5061/dryad.8682 | Available at Dryad Digital Repository under a CC0 Public Domain Dedication |

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
