## [Decision Letter]

Thank you for submitting your article "Personality links with lifespan in chimpanzees" for consideration by *eLife*. Your article has been reviewed by three peer reviewers, one of whom is a member of our Board of Reviewing Editors, and the evaluation has been overseen by Ian Baldwin as the Senior Editor. The reviewers have opted to remain anonymous.

The reviewers have discussed the reviews with one another and the Reviewing Editor has drafted this decision to help you prepare a revised submission.

Summary:

The authors report a study that links personality with lifespan of a large sample of captive chimpanzees. They rely on a questionnaire method for assessing personality along six dimensions, and they use those data to find a relationship between personality and longevity. They find that for males longevity is related to agreeableness and for females to openness, and they discuss these results in terms of phylogenetic assumptions about human and great ape evolution.

Essential revisions:

Whereas the results are interesting, the reviewers and Reviewing Editor all have several major concerns with this study that cast doubt on the robustness of the theoretical framing, the methods, and as a consequence, the conclusions of the paper. A major revision would be required in order for publication in *eLife*, but as most of these involve incorporation of key discussion, literature, and modifications to the analytical approach, we feel these are feasible to accomplish within a two-month time frame. If you choose to revise and resubmit along the lines below, this will be a much stronger paper, and can be considered for publication in *eLife*. If these revisions are undertaken, it will have the potential to contribute not only an important dataset, but also to address some key theoretical problems in this area of research more broadly.

Below are the major issues/points that must be addressed in order to be considered in revised form. Large sections from the original reviews have been pasted into these comments, as they contain many helpful suggestions about theoretical framing, literature, and approach. Although each point is described at length, the substance of the revision should not simply be a longer version of the existing manuscript. Instead, it should be only a slightly longer version that revises much of the background and discussion in light of the valuable insights from the reviewers below.

1) Title. Given some of the uncertainties in the data and conclusions that are raised below, the title should be modified to ensure it faithfully reflects these issues.

2) Phylogenetic assumptions. The central framing of the problem with respect to human evolution (Introduction, fourth paragraph) makes a standard phylogenetic assumption: that behavioral attributes found in both chimpanzees and humans should also have been shared by the last common ancestor. There is much debate over the utility of this approach for behavioral attributes, or if it might suffer overmuch from homoplasy – especially when there appear to be so many differences between chimpanzees and bonobos, with both equally related to humans (work by Sayers is relevant here). The authors should include discussion and literature that explains why here they make the claim so strongly, or why they claim to pinpoint the evolution of specific male and female personality attributes to such a specific period of time (since the split with hominins). This is especially the case when their research question (as currently states) prioritizes similarities between humans and chimpanzees (rather than bonobos). The discussion that leads to this point also appears to somewhat answer the question in advance, because it is clear that the authors are already arguing that a link between personality and fitness is ancestral in primates (and therefore will also be linked in chimpanzees, humans, presumably bonobos, and their last common ancestor). This argument seeps back in later with the discussion of gorillas (Discussion, third paragraph), where they are referred to rather oddly (considering the phylogenetic argument up to this point and the actual genetic relatedness of gorillas and chimps versus chimps and bonobos or chimps and humans) as "close chimpanzee cousins". Thus, the phylogenetic argument should be very carefully constructed: what traits do they think are derived in chimpanzees, what traits ancestral to both chimpanzees and humans, what is the rationale, etc.?

3) Use of literature. The paper opens with a strong background discussion, but reads like a bank of examples rather than a coherent lead-up to a clear set of hypotheses (more on this below). Each cited study shows a suggestive tendency for proxies of fitness to link with some aspect of personality, but there is clear diversity within primates regarding how each of these relationships actually plays out (as well as in what fitness measures are used). This background captures some of the ambiguity and gaps in current knowledge, but it does not confront them head-on. The paper would increase its impact if it clearly describes the areas where there is more surety than others, and specifically where increased work is necessary and why. There is also a near-complete lack of citations to the non-primate animal personality literature. It is generally disconcerting that the primate personality and non-primate personality literatures don't often cite each other, but it is especially puzzling here since researchers working with short-lived, easily manipulated species like birds, fish, or insects, have much better means of studying links between behavioral differences and fitness outcomes, and can add a solid empirical basis to the theoretical framework needed in this study (see below).

4) Questionnaire method. There are some serious concerns about the questionnaire approach to assessing personality, but we recognize this is an approach that is used by some researchers and that the justification simply must be more robustly presented. Perhaps what is most problematic is that in this manuscript, the authors present this method as *the* method, thereby ignoring an enormous field of (ecological) studies that instead use an ethological approach and code actual behavior, or conduct experiments to test for consistency of behavior across time and context. Additionally, this field has proposed actual informed hypotheses about how personality influences life-history traits. The fact that the authors do not even refer to any of the work done by, to name only a few, Dingemanse, Réale and Sih, while talking about animal personality in light of evolution, is extremely problematic. Also, such studies do exist for chimpanzees; e.g. Uher and Asendorf 2008; Koski 2011; Massen et al., 2013, and also this should be acknowledged. Personality is defined as "inter-individual differences that are consistent over time and context" (something that the authors do not mention), and whereas they report here on inter-individual differences, they do not report anything about it being consistent or repeatable. It is common (and good) practice to use a test-retest design to check for such consistency, yet in primatology and especially when testing apes, this seems to be deemed unnecessary. Yet, as mentioned, this is how animal personality is defined and thus is very important. These authors (and others) tend to use the high inter-rater reliability they find as an argument against this. However, this is not the same as temporal consistency, and as mentioned before, this inter-rater reliability doesn't result from independent raters. Zoo keepers (and researchers alike) talk about their animals, and thus inadvertently but unavoidably, influence each other’s perception of, and consequently the ratings of, these animals. Further, all the references the authors use to validate the use of the rating method in the Materials and methods section (Weiss et al., 2009; King and Figueredo, 1997; Weiss, King and Hopkins, 2007; Herrelko, Vick and Buchanan-Smith, 2012; Vazine et al., 2007; Weiss, King and Figueredo, 2000; Wilson et al., 2017; Latzman et al., 2015; King, Weiss and Farmer, 2005; Weiss et al., 2012; Pederson, King and Landau, 2005; Latzman et al., 2015; Blatchley and Hopkins, 2010 –) are of people involved in this study, and thus not independent. In short, serious discussion needs to be undertaken to ensure the reader is aware that the questionnaire approach has its detractors, and the basis of those critiques. This has the potential to make this a much stronger paper because by providing a balanced view it can simultaneously present new data and pre-empt the problem of citation divergence (whereby some research groups cite only from select literature and others from a different set, and thus integration of these two literatures becomes compromised and thus detrimental to the overall scientific aims).

5) Hypothesis testing. The hypotheses are not clearly set up from the start, they are not embedded in any relevant theory (more on this below), and they are post hoc in nature. It appears to be a study where a large sample was input into some analyses to see what patterns emerged, and then those patterns were explained after the fact. A better approach would be to structure the lead-up so that it is clear what would be expected under what circumstances (phylogenetic, environmental, life history, etc.) and then test those hypotheses. In addition to modifying the setup to create a more rigorous set of well-supported expectations, the question be reworded to be more specific about chimpanzees and humans with respect to what traits should be linked with longevity. There is a good start to this discussion in the Introduction, and that could form the basis for a revised setup to the problem. There appear to be some expectations (Discussion, second paragraph), and these are discussed in an interesting way later, but the manuscript would be much stronger if these were clearly defined at the start and then systematically tested.

6) Theoretical framing. All predictions or interpretations are entirely based on previous empirical results, rather than derived from first principles. While this might be the norm in psychology, *eLife* is a biology journal, and evolutionary theory provides us with a framework from which to derive predictions about biological traits such as longevity and stable behavioral variation (that is presumably mediated by stable variation in neurobiology, metabolism, etc.). Thus, when examining links between consistent behavioral differences and longevity, an evolutionary biologist immediately thinks of life-history strategy as a possible underlying cause of both. Life history theory is especially relevant here as all sources of extrinsic mortality have been removed in this captive sample, and the chimpanzees are presumably dying because of intrinsic mortality; an individual's degree of investment in maintenance and repair, i.e. the things that reduce intrinsic mortality, is of course shaped by their life-history strategy, as are, arguably, consistent behavioral differences between individuals. Indeed, there is a large literature examining links between personality, longevity, and measures to reduce intrinsic mortality such as investment in immune function, in other animals called the 'pace-of-life syndrome' (e.g. Reale et al., 2010; Smith and Blumstein, 2008). This literature provides the kind of a priori predictions the current manuscript is lacking, such as certain personality dimensions being linked to longevity due to being part of a faster or slower life-history strategy. For example, achieving high dominance requires substantial investment in physical strength and muscle, associated with high testosterone levels and risk-taking behavior, which trade-off with investment in immune function etc. and are thus associated with a faster life-history strategy and higher extrinsic and intrinsic mortality (as is typical for most primate males compared to females); hence one would predict dominance to be negatively associated with longevity not positively, similar to the general sex difference in dominance and longevity (see e.g. Kruger and Nesse, 2005, Human Nature An evolutionary life history framework for understanding sex differences in human mortality rates). Conversely, an association between agreeableness and longevity is exactly what you would predict if agreeable individuals invest less in behavioral dominance and more in cooperation, which would be associated with a slower life-history strategy. Throughout the paper the authors speculate about possible causal links between personality traits and longevity (through 'controlling health', or 'health benefits conferred by intelligence'), which will need to be re-examined in light of theory that predicts both to be explained by a third variable (life-history strategy). For relevant arguments in humans, see e.g. several articles by Pepper and Nettle (2014 Human Nature, 2014 Applied evolutionary anthropology, 2017 Behavioral and Brain Sciences) that argue how a life history theory perspective can help explain variation in health behavior and thus SES-gradients in health. Of course, there are other evolutionary theories of personality (see e.g. Buss, 2009 How can evolutionary psychology successfully explain personality and individual differences?) but life-history theory provides the most direct link to longevity.

7) Use of a captive sample. The authors make strong claims about evolution and *natural* selection, yet test animals in a (non-natural) captive situation. As a consequence, selection pressures that have shaped evolution are being cancelled out and the effects of personality on longevity that the authors report are not informative for understanding the evolution of chimpanzees. For example, in this study there is no effect of extraversion (or boldness) on longevity, but it is obvious that such a trait may have an effect with actual predators around. Similarly, in the wild, were food is a limiting factor, dominance (which may not actually be a personality trait as it is not consistent if new opportunities arise) will have a major effect. As another example, the authors simultaneously argue that "observed effects in captive chimpanzees will be more comparable to effects found in similar human studies than would effects observed in wild chimpanzees". However, they go on to then offer an evolutionary explanation that seeks to describe their results in terms of ancestral behavior and the environment of selection (Abstract): "natural selection, after the divergence of hominins, favored the protective effects of high quality social bonds for males and exploratory behavior for females." The relationships they observe in fact seem equally explicable as factors that promote longevity specifically in captive situations. Although the authors do well to note this possibility, they appear to dismiss it in favor of their preferred alternative. Where they do find a lack of concordance with their expectations, the authors quickly engage in a useful discussion about the effects of captivity, but seem to discard this argument when they discuss their positive results. These alternative explanations must be carefully explored, and test implications set out (with substantive literature support) in order to seriously treat (and not just dismiss) the very real possibility that the observed pattern has no bearing at all on natural selection. One reviewer note that the captive sample can have its advantages, and these can be stressed. For example, the captive sample eliminates most extrinsic mortality, so that what remains is essentially how much individuals invest in maintenance and repair, which could well be related to their personality through life-history strategy (slow strategy = invest more = lower intrinsic mortality = 'nicer' personality). This still suffers from the problem that extrinsic mortality matters a lot in wild populations (and thus natural selection), but acknowledging these shortcomings, this study could be a good test of the idea that life-history strategy has consequences for both behavioral style and intrinsic mortality risk.

8) Analytical approach.a) It appears that the power analyses were conducted on the entire dataset (rather than pilot data), and thus constitute 'observed power'; this is unfortunately completely flawed and unnecessary. As demonstrated by Hoenig and Heisey (2001, Am Stat The abuse of power: The pervasive fallacy of power calculations for data analysis) there is nothing to be gained from such a retrospective power analysis, and indeed they may be entirely misleading. Power analysis only makes sense prospectively, using pilot data, and indeed *eLife*'s transparent reporting form asks 'whether an appropriate sample size was computed *when the study was being designed*'. As this was not the case here, the power analyses should be removed.

b) I have to disagree with the dismissal of an age-confound on agreeableness based on a non-significant P-value of 0.077. P value thresholds are arbitrary conventions, and when there is an age pattern – the correlation of -0.08 is about as strong as the one for neuroticism at 0.09 – it should be controlled for, *especially* when one of the main findings is about an association between agreeableness and longevity. And while I appreciate that the authors fit several possible age models to the personality dimensions that did have significant correlations with age, I also disagree with selecting a single best model based on AICc (as the authors know, information criteria are better used to weight models and average predictions rather than select a single model [unless it receives all the weight]). Furthermore, polynomials are not ideal, and I would suggest using a spline term for age (using GAM) instead, which obviates the need to compare linear vs. non-linear fits. Incidentally, the fact that the best fit for the age effect on most personality dimensions was non-linear refutes the use of simple correlations. I would thus strongly suggest using GAM residuals for each personality dimension. As an aside, I was confused as to why date of birth rather than biological age was used?

9) Data accessibility. Two of the reviewers also expressed concern that the entire dataset may not be de-identified and available in published form (by assigning an ID to individual chimpanzees, and facilities). The editorial staff also had a discussion about the submission's compliance with the open-access policy of *eLife*. It is not clear to what extent the data can be precisely reproduced, given a lack of access to the full dataset that was used in the analysis. For example, how can personality links with lifespan be replicated without mortality data for the same individuals for which the personality attributes are known? Knowing social relationships, group size, etc. are also important because certain personality traits may be much more important in some specific settings than others.

---

## [Author Response]

1) Title. Given some of the uncertainties in the data and conclusions that are raised below, the title should be modified to ensure it faithfully reflects these issues.

Were-analyzed our data (discussed below), and these changes did not drastically change our findings. Since the results are unchanged, we do not see an obvious way to improve the title, but welcome the reviewers’ thoughts on this.

2) Phylogenetic assumptions. The central framing of the problem with respect to human evolution (Introduction, fourth paragraph) makes a standard phylogenetic assumption: that behavioral attributes found in both chimpanzees and humans should also have been shared by the last common ancestor. There is much debate over the utility of this approach for behavioral attributes, or if it might suffer overmuch from homoplasy – especially when there appear to be so many differences between chimpanzees and bonobos, with both equally related to humans (work by Sayers is relevant here). The authors should include discussion and literature that explains why here they make the claim so strongly, or why they claim to pinpoint the evolution of specific male and female personality attributes to such a specific period of time (since the split with hominins). This is especially the case when their research question (as currently states) prioritizes similarities between humans and chimpanzees (rather than bonobos).

We do not wish to understate how informative studies of bonobos are of these same issues; unfortunately, data from bonobos regarding longevity, are lacking. We have revised our text to acknowledge the importance of bonobos in this framework, though again, there is unfortunately little evidence from bonobos that we can bring to bear on these issues.

We have also addressed this concern in our addition of a phylogenetic perspective on great ape personality dimensions. To expand on our descriptions in the text, gorillas possess dominance, extraversion, and agreeableness (1), as well as dimensions like openness (2) and conscientiousness (3), but the evidence for these is weaker. Humans and chimpanzees possess the same Big 5 factors, but chimpanzees have the additional dominance factor (4-6). Bonobos differ from chimpanzees, humans, and possibly gorillas, as bonobos have a dominance-like factors (“assertiveness”), two factors describing different aspects of conscientiousness, an openness and agreeableness factor, and an extraversion factor that may also be termed “social withdrawal” (7).

From this, we can say that the most phylogenetically parsimonious tracing of personality in African apes is that dominance (or assertiveness), extraversion, agreeableness, neuroticism, openness, and possibly conscientiousness, are ancestral in African apes. The bonobo configuration regarding extraversion, conscientiousness, and neuroticism is derived, as is the lack of dominance in humans. A concise version of this explanation is now given in the manuscript (–Introduction, ninth paragraph).

The discussion that leads to this point also appears to somewhat answer the question in advance, because it is clear that the authors are already arguing that a link between personality and fitness is ancestral in primates (and therefore will also be linked in chimpanzees, humans, presumably bonobos, and their last common ancestor). This argument seeps back in later with the discussion of gorillas (Discussion, third paragraph), where they are referred to rather oddly (considering the phylogenetic argument up to this point and the actual genetic relatedness of gorillas and chimps versus chimps and bonobos or chimps and humans) as "close chimpanzee cousins". Thus, the phylogenetic argument should be very carefully constructed: what traits do they think are derived in chimpanzees, what traits ancestral to both chimpanzees and humans, what is the rationale, etc.?

Please see our previous response. We have revised our review of the literature, and in particular we have expanded our theory that explains our arguments as to which personality-longevity associations may be ancestral or derived. Please see our responses to query 5 concerning hypothesis testing, as well.

3) Use of literature. The paper opens with a strong background discussion, but reads like a bank of examples rather than a coherent lead-up to a clear set of hypotheses (more on this below). Each cited study shows a suggestive tendency for proxies of fitness to link with some aspect of personality, but there is clear diversity within primates regarding how each of these relationships actually plays out (as well as in what fitness measures are used). This background captures some of the ambiguity and gaps in current knowledge, but it does not confront them head-on. The paper would increase its impact if it clearly describes the areas where there is more surety than others, and specifically where increased work is necessary and why.

We have revised our Introduction and Discussion with an eye on confronting the strengths, weaknesses, and applicability of the related bodies of literature in primate behavior, psychology, and behavioural ecology. In doing so, we incorporated life-history theory, while also attempting to reconcile the nonhuman primate research with the different approaches and results in humans. We have attempted to address the main theoretical and empirical gaps in the existing literature, while maintaining a cohesive picture of the broader theory.

There is also a near-complete lack of citations to the non-primate animal personality literature. It is generally disconcerting that the primate personality and non-primate personality literatures don't often cite each other, but it is especially puzzling here since researchers working with short-lived, easily manipulated species like birds, fish, or insects, have much better means of studying links between behavioral differences and fitness outcomes, and can add a solid empirical basis to the theoretical framework needed in this study (see below).

We agree with the reviewers that better integration of the literatures should be a goal of such studies and regret that we did not set a good example in the previous version of this manuscript. To address this, we expanded our Introduction to recognize and incorporate relevant reviews and syntheses, which are largely based on non-primates. Moreover, where possible and appropriate, we have cited specific studies from non-primates.

4) Questionnaire method. There are some serious concerns about the questionnaire approach to assessing personality, but we recognize this is an approach that is used by some researchers and that the justification simply must be more robustly presented. Perhaps what is most problematic is that in this manuscript, the authors present this method as the method, thereby ignoring an enormous field of (ecological) studies that instead use an ethological approach and code actual behavior, or conduct experiments to test for consistency of behavior across time and context. Additionally, this field has proposed actual informed hypotheses about how personality influences life-history traits. The fact that the authors do not even refer to any of the work done by, to name only a few, Dingemanse, Réale and Sih, while talking about animal personality in light of evolution, is extremely problematic. Also, such studies do exist for chimpanzees; e.g Uher and Asendorf 2008; Koski 2011; Massen et al., 2013, and also this should be acknowledged.

Our original intention was to describe the history and validity of the method we used, but we apparently did so at the cost of discussing the broader issues. To address this weakness in the manuscript, we expanded the background of our methods to describe related methods including those that do not use questionnaires. We have done this in the current version, citing (to the best of our knowledge) all personality studies that have examined chimpanzees, using either subjective ratings or behavioral measures. This includes all the papers mentioned by the reviewers, as well as others. We also incorporated the foundational work of non-primatologist behavioural ecologists who study personality, Réale and Dingemanse in particular.

We believe this comprehensive take on the extant literature has improved the manuscript, as it demonstrates the convergence of personality measures across studies and approaches. We mention this in the paper (–Introduction, ninth and tenth paragraphs), but we wish to point it out here as well, that our questionnaire is more than just a set of adjectives – it includes both adjectives and a behavioural description for each adjective.

Personality is defined as "inter-individual differences that are consistent over time and context" (something that the authors do not mention), and whereas they report here on inter-individual differences, they do not report anything about it being consistent or repeatable. It is common (and good) practice to use a test-retest design to check for such consistency, yet in primatology and especially when testing apes, this seems to be deemed unnecessary. Yet, as mentioned, this is how animal personality is defined and thus is very important. These authors (and others) tend to use the high inter-rater reliability they find as an argument against this. However, this is not the same as temporal consistency, and as mentioned before, this inter-rater reliability doesn't result from independent raters. Zoo keepers (and researchers alike) talk about their animals, and thus inadvertently but unavoidably, influence each other’s perception of, and consequently the ratings of, these animals.

Before responding, we wish to correct a misperception. There is an understanding of the importance of repeatability as one way to establish the reliability of personality measures in the primate literature. We are unaware of anybody who argues that this is not important or who use interrater reliabilities as an argument against also establishing repeatabilities. In fact, repeatabilities or re-test reliabilities of personalities in primates have been examined since at least the 1970s, and this has also played a large role in studies of human personality research. However, given the origins of these subjects, it is not always possible to obtain these kinds of data (keepers often have limited time to participate).

Nevertheless, in our expanded Introduction, we discuss the repeatability of measures (tenth paragraph). Several studies (some involving a subset of our data and some independent studies) have examined temporal consistency in chimpanzee personality, occasionally over the course of decades (6, 8-10). In their review of the primate personality literature, Freeman and Gosling (11) note six studies, of rhesus macaques and the great apes, that presented test-retest reliability. Since that review, more papers have demonstrated repeatability (12-15).

Further, all the references the authors use to validate the use of the rating method in the Materials and methods section (Weiss et al., 2009; King and Figueredo, 1997; Weiss, King and Hopkins, 2007; Herrelko, Vick and Buchanan-Smith, 2012; Vazine et al., 2007; Weiss, King and Figueredo, 2000; Wilson et al., 2017; Latzman et al., 2015; King, Weiss and Farmer, 2005; Weiss et al., 2012; Pederson, King and Landau, 2005; Latzman et al., 2015; Blatchley and Hopkins, 2010) are of people involved in this study, and thus not independent. In short, serious discussion needs to be undertaken to ensure the reader is aware that the questionnaire approach has its detractors, and the basis of those critiques. This has the potential to make this a much stronger paper because by providing a balanced view it can simultaneously present new data and pre-empt the problem of citation divergence (whereby some research groups cite only from select literature and others from a different set, and thus integration of these two literatures becomes compromised and thus detrimental to the overall scientific aims).

Initially, we took this approach because other ratings studies were most comparable to our in terms of methods used, applications, etc. As mentioned above, we have now cited the chimpanzee (and other primate) literature more widely. In our Introduction we discuss the criticisms of rating techniques, and mention reliability and validity of our instrument in both the Introduction and methods. Ultimately, the revised manuscript gives what we hope is a balanced take on the strengths of the approach and the convergent validity of these methods for assessing personality in chimpanzees.

5) Hypothesis testing. The hypotheses are not clearly set up from the start, they are not embedded in any relevant theory (more on this below), and they are post hoc in nature. It appears to be a study where a large sample was input into some analyses to see what patterns emerged, and then those patterns were explained after the fact. A better approach would be to structure the lead-up so that it is clear what would be expected under what circumstances (phylogenetic, environmental, life history, etc.) and then test those hypotheses. In addition to modifying the setup to create a more rigorous set of well-supported expectations, the question be reworded to be more specific about chimpanzees and humans with respect to what traits should be linked with longevity. There is a good start to this discussion in the Introduction, and that could form the basis for a revised setup to the problem. There appear to be some expectations (Discussion, second paragraph), and these are discussed in an interesting way later, but the manuscript would be much stronger if these were clearly defined at the start and then systematically tested.

We have rewritten the entire Introduction to incorporate life history theory into our manuscript, as the central, guiding theory. From the outset, we develop the theory of personality in that context, before discussing specifics of primates and personality measurement, and finally situating our hypotheses in the life history framework.

6) Theoretical framing. All predictions or interpretations are entirely based on previous empirical results, rather than derived from first principles. While this might be the norm in psychology, eLife is a biology journal, and evolutionary theory provides us with a framework from which to derive predictions about biological traits such as longevity and stable behavioral variation (that is presumably mediated by stable variation in neurobiology, metabolism, etc.). […] Throughout the paper the authors speculate about possible causal links between personality traits and longevity (through 'controlling health', or 'health benefits conferred by intelligence'), which will need to be re-examined in light of theory that predicts both to be explained by a third variable (life-history strategy). For relevant arguments in humans, see e.g. several articles by Pepper and Nettle (2014 Human Nature, 2014 Applied evolutionary anthropology, 2017 Behavioral and Brain Sciences) that argue how a life history theory perspective can help explain variation in health behavior and thus SES-gradients in health. Of course, there are other evolutionary theories of personality (see e.g. Buss, 2009 How can evolutionary psychology successfully explain personality and individual differences?) but life-history theory provides the most direct link to longevity.

As mentioned in the previous response, we have rewritten the paper, from the Introduction onward, with life history theory at the front and centre. We believe that our adherence to the theory and the predictions we draw from it accurately reflect the extrinsic and intrinsic factors that are in play in this population.

7) Use of a captive sample. The authors make strong claims about evolution and natural selection, yet test animals in a (non-natural) captive situation. As a consequence, selection pressures that have shaped evolution are being cancelled out and the effects of personality on longevity that the authors report are not informative for understanding the evolution of chimpanzees. For example, in this study there is no effect of extraversion (or boldness) on longevity, but it is obvious that such a trait may have an effect with actual predators around. Similarly, in the wild, were food is a limiting factor, dominance (which may not actually be a personality trait as it is not consistent if new opportunities arise) will have a major effect. As another example, the authors simultaneously argue that "observed effects in captive chimpanzees will be more comparable to effects found in similar human studies than would effects observed in wild chimpanzees". However, they go on to then offer an evolutionary explanation that seeks to describe their results in terms of ancestral behavior and the environment of selection (Abstract): "natural selection, after the divergence of hominins, favored the protective effects of high quality social bonds for males and exploratory behavior for females." The relationships they observe in fact seem equally explicable as factors that promote longevity specifically in captive situations. Although the authors do well to note this possibility, they appear to dismiss it in favor of their preferred alternative. Where they do find a lack of concordance with their expectations, the authors quickly engage in a useful discussion about the effects of captivity, but seem to discard this argument when they discuss their positive results. These alternative explanations must be carefully explored, and test implications set out (with substantive literature support) in order to seriously treat (and not just dismiss) the very real possibility that the observed pattern has no bearing at all on natural selection. One reviewer note that the captive sample can have its advantages, and these can be stressed. For example, the captive sample eliminates most extrinsic mortality, so that what remains is essentially how much individuals invest in maintenance and repair, which could well be related to their personality through life-history strategy (slow strategy = invest more = lower intrinsic mortality = 'nicer' personality). This still suffers from the problem that extrinsic mortality matters a lot in wild populations (and thus natural selection), but acknowledging these shortcomings, this study could be a good test of the idea that life-history strategy has consequences for both behavioral style and intrinsic mortality risk.

We have scaled back our claims about the explanatory power of this sample, and have reoriented our findings in the context of the extrinsic and intrinsic factors, relevant to each personality dimension, that are or are not in play for captive chimpanzees. In our revised hypotheses, we consider all the major extrinsic and intrinsic differences between captive and wild environments, and how these would be related to strategic life-history differences. We also highlighted the drawbacks and advantages to working with a captive sample (–Introduction, eleventh paragraph; Discussion, seventh paragraph).

8) Analytical approach.a) It appears that the power analyses were conducted on the entire dataset (rather than pilot data), and thus constitute 'observed power'; this is unfortunately completely flawed and unnecessary. As demonstrated by Hoenig and Heisey (2001, Am Stat The abuse of power: The pervasive fallacy of power calculations for data analysis) there is nothing to be gained from such a retrospective power analysis, and indeed they may be entirely misleading. Power analysis only makes sense prospectively, using pilot data, and indeed eLife's transparent reporting form asks 'whether an appropriate sample size was computed when the study was being designed'. As this was not the case here, the power analyses should be removed.

We recognize the issues with power analysis in the context of our study. We hoped the analyses would be useful to a statistically literate audience, but since the reviewers point out that the analyses could be misleading or distracting, we have removed them.

b) I have to disagree with the dismissal of an age-confound on agreeableness based on a non-significant P-value of 0.077. P value thresholds are arbitrary conventions, and when there is an age pattern – the correlation of -0.08 is about as strong as the one for neuroticism at 0.09 – it should be controlled for, especially when one of the main findings is about an association between agreeableness and longevity. And while I appreciate that the authors fit several possible age models to the personality dimensions that did have significant correlations with age, I also disagree with selecting a single best model based on AICc (as the authors know, information criteria are better used to weight models and average predictions rather than select a single model [unless it receives all the weight]). Furthermore, polynomials are not ideal, and I would suggest using a spline term for age (using GAM) instead, which obviates the need to compare linear vs. non-linear fits. Incidentally, the fact that the best fit for the age effect on most personality dimensions was non-linear refutes the use of simple correlations. I would thus strongly suggest using GAM residuals for each personality dimension. As an aside, I was confused as to why date of birth rather than biological age was used?

We very much appreciate the suggestion that we used GAMs to control for age confounding in the personality data. We were not fully aware of the applicability of this method to the problem we faced. In response, we have now reanalysed the age and personality relationships for all traits in a GAM framework, and have used the GAM residuals in subsequent analyses instead of the simpler linear and polynomial model residuals. We have also changed the confounding variable from DoB to age at personality rating. These changes did not alter our results, except the association between longevity and residualized openness in females. The hazard ratio shifted by about 0.05, and the new 95% confidence interval now overlaps with 0. We have scaled back our conclusions accordingly, and our revised manuscript interprets openness and longevity in light of the ambiguity of this finding (Discussion, fourth paragraph).

9) Data accessibility. Two of the reviewers also expressed concern that the entire dataset may not be de-identified and available in published form (by assigning an ID to individual chimpanzees, and facilities). The editorial staff also had a

Since submitting the paper we have consulted with all authors and interested parties on the matter of data accessibility. We are pleased to say that all the data we used in our analyses can be published along with the manuscript, should it be accepted.

References

1) Gold KC, Maple TL. Personality assessment in the gorilla and its utility as a management tool. Zoo Biol. 1994;13(5):509-22.

2) Eckardt W, Steklis HD, Steklis NG, Fletcher AW, Stoinski TS, Weiss A. Personality dimensions and their behavioral correlates in wild Virunga mountain gorillas (Gorilla beringei beringei). Journal of Comparative Psychology. 2015;129(1):26-41.

3) Schaefer SA, Steklis HD. Personality and subjective well‐being in captive male western lowland gorillas living in bachelor groups. Am J Primatol. 2014;76:879-89.

4) King JE, Figueredo AJ. The Five-Factor Model plus Dominance in chimpanzee personality. Journal of Research in Personality. 1997;31:257-71.

5) Freeman HD, Brosnan SF, Hopper LM, Lambeth SP, Schapiro SJ, Gosling SD. Developing a comprehensive and comparative questionnaire for measuring personality in chimpanzees using a simultaneous top-down/bottom-up design. Am J Primatol. 2013;75(10):1042-53.

6) Dutton DM. Subjective assessment of chimpanzee (Pan troglodytes) personality: Reliability and stability of trait ratings. Primates. 2008;49:253-9.

7) Weiss A, Staes N, Pereboom JJM, Inoue-Murayama M, Stevens JMG, Eens M. Personality in bonobos. Psychol Sci. 2015;26(9):1430-9.

8) King JE, Weiss A, Sisco MM. Aping humans: Age and sex effects in chimpanzee (Pan troglodytes) and human (Homo sapiens) personality. Journal of Comparative Psychology. 2008;122:418-27.

9) Buirski P, Plutchik R, Kellerman H. Sex differences, dominance, and personality in the chimpanzee. Anim Behav. 1978;26:123-9.

10) Weiss A, Wilson ML, Collins DA, Mjungu D, Kamenya S, Foerster S, et al. Personality in the chimpanzees of Gombe National Park. Scientific data. 2017;4:170146.

11) Freeman HD, Gosling SD. Personality in nonhuman primates: A review and evaluation of past research. Am J Primatol. 2010;72:653-71.

12) Weiss A, Adams MJ, Widdig A, Gerald MS. Rhesus macaques (Macaca mulatta) as living fossils of hominoid personality and subjective well-being. Journal of Comparative Psychology. 2011;125:72-83.

13) von Borell C, Kulik L, Widdig A. Growing into the self: the development of personality in rhesus macaques. Anim Behav. 2016;122:183-95.

14) Neumann C, Agil M, Widdig A, Engelhardt A. Personality of wild male crested macaques (Macaca nigra). PLoS ONE. 2013;8:e69383.

15) Brent LJN, Semple S, MacLarnon A, Ruiz-Lambides A, Gonzalez-Martinez J, Platt MJ. Personality traits in rhesus macaques (Macaca mulatta) are heritable but do not predict reproductive output. Int J Primatol. 2014;35:188-209.